# Oral Administration of Protease-Soluble Chicken Type II Collagen Ameliorates Anterior Cruciate Ligament Transection–Induced Osteoarthritis in Rats

**DOI:** 10.3390/nu15163589

**Published:** 2023-08-16

**Authors:** Nan-Fu Chen, Yen-You Lin, Zhi-Kang Yao, Chung-Chih Tseng, Yu-Wei Liu, Ya-Ping Hung, Yen-Hsuan Jean, Zhi-Hong Wen

**Affiliations:** 1Division of Neurosurgery, Department of Surgery, Kaohsiung Armed Forces General Hospital, Kaohsiung 80284, Taiwan; g1078020008@mail.802.org.tw; 2Institute of Medical Science and Technology, National Sun Yat-sen University, Kaohsiung 80424, Taiwan; caviton@g-mail.nsysu.edu.tw; 3Department of Sports Medicine, China Medical University, Taichung 40402, Taiwan; chas6119@mail.cmu.edu.tw; 4Department of Orthopedics, Kaohsiung Veterans General Hospital, Kaohsiung 81362, Taiwan; akang329@vghks.gov.tw; 5Department of Marine Biotechnology and Resources, National Sun Yat-sen University, Kaohsiung 80424, Taiwan; yoweiwei@g-mail.nsysu.edu.tw; 6R&D Department, Taiyen Biotech Co., Ltd., Tainan 70263, Taiwan; tsi201@tybio.com.tw; 7Department of Orthopedic Surgery, Pingtung Christian Hospital, Pingtung 90059, Taiwan; 8Institute of BioPharmaceutical Sciences, National Sun Yat-sen University, Kaohsiung 80424, Taiwan

**Keywords:** anterior cruciate ligament transection, type II collagen, protease, nociception, cartilage, inflammation

## Abstract

This study investigated whether oral supplementation with protease-soluble chicken type II collagen (PSCC-II) mitigates the progression of anterior cruciate ligament transection (ACLT)–induced osteoarthritis (OA) in rats. Eight-week-old male Wistar rats were randomly assigned to the following groups: control, sham, ACLT, group A (ACLT + pepsin-soluble collagen type II collagen (C-II) with type I collagen), group B (ACLT + Amano M–soluble C-II with type I collagen), group C (ACLT + high-dose Amano M–soluble C-II with type I collagen), and group D (ACLT + unproteolyzed C-II). Various methods were employed to analyze the knee joint: nociceptive tests, microcomputed tomography, histopathology, and immunohistochemistry. Rats treated with any form of C-II had significant reductions in pain sensitivity and cartilage degradation. Groups that received PSCC-II treatment effectively mitigated the ACLT-induced effects of OA concerning cancellous bone volume, trabecular number, and trabecular separation compared with the ACLT alone group. Furthermore, PSCC-II and unproteolyzed C-II suppressed ACLT-induced effects, such as the downregulation of C-II and upregulation of matrix metalloproteinase-13, tumor necrosis factor-α, and interleukin-1β. These results indicate that PSCC-II treatment retains the protective effects of traditional undenatured C-II and provide superior benefits for OA management. These benefits encompass pain relief, anti-inflammatory effects, and the protection of cartilage and cancellous bone.

## 1. Introduction

Osteoarthritis (OA) is the most common disorder of the musculoskeletal system and a leading cause of joint dysfunction and disability worldwide [1]. OA is generally caused by aging or mechanical-induced dysfunction of biological factors, resulting in an imbalance in cartilage homeostasis. This imbalance causes the degradation of the extracellular matrix (ECM), hyaluronic acid, and proteoglycans in cartilage tissue. It also leads to fibrillation and erosion of the articular surface, chondrocyte death, matrix calcification, and vascular invasion [2]. Excluding the water content, the ECM of articular cartilage comprises collagen (60%), proteoglycans (25–35%), and other noncollagenous proteins (15–20%). Type II collagen (C-II) accounts for 80% of the total collagen and provides mechanical stability to cartilage [3]. Without timely supplementation with nutraceuticals, excessive collagen degradation by matrix metalloproteinases (MMPs) can damage cartilage tissue [4,5].

The current primary treatment for OA involves reducing inflammation, alleviating pain and discomfort, and improving the structure of collagen or temporarily delaying the progression of the disease [6]. Recently, many studies have investigated whether supplementation with nutraceuticals containing undenatured C-II can alleviate OA progression [7]. In a retrospective 2020 study, Hasan et al. found that undenatured C-II improved synovitis and cartilage degradation in humans, horses, dogs, and rodents with OA [8]. Chicken sternal cartilage is a common source for preparing undenatured C-II. In traditional undenatured C-II preparations, the triple helix structure of collagen is retained, and it resembles that of C-II in human cartilage tissue. However, one study reported that its high molecular weight may result in poor absorption [9]. The processing of undenatured C-II ensures the preservation of protein glycosylation and telopeptides, which may participate in immune responses [8,10]. Studies have revealed that native glycosylation on C-II can cause the excessive activation of T cells [11,12]. Degrading or modifying the glycosylation on C-II significantly reduced the incidence, time of onset, and severity of C-II-induced rheumatoid arthritis (RA) in mouse models [11,12]. Moreover, clinical studies have revealed that T cells exhibited a strong response to C-II in patients with RA, and this immune response was correlated with disease severity [13,14,15]. These results highlight the risk of an excessive immune response to the structural components (e.g., carbohydrates or telopeptides) of native C-II, which may increase joint inflammation.

In cartilage tissue, type IX collagen is linked to C-II through polysaccharide cross-linking, forming a robust fibril structure [16]. Therefore, proteases such as pepsin can degrade the polysaccharide side chains on collagen, increasing the release of C-II from the fibril structure and enhancing the extraction efficiency [17]. Removing the telopeptide region from C-II through protease cleavage can increase water solubility and potentially reduce immune responses [18]. Although undenatured C-II has oral tolerance properties, which are less than the threshold of immune response activation, preserved post-translational modifications or structures may trigger adverse immune reactions in the human body [19,20]. The addition of proteases during the extraction process can disrupt the interaction between type IX collagen and C-II. It can reduce the immune response risks associated with the natural structure of C-II. The present study primarily investigated the protective effects of undenatured C-II obtained through protease addition during the extraction of chicken sternal cartilage on rats with OA.

## 2. Materials and Methods

### 2.1. Preparation of Protease-Soluble C-II

Cleaned chicken sternal cartilage without adhering tissue was donated by Taiyen Biotech in Tainan, Taiwan. The cartilage was cut into small pieces (1–2 mm^3^) and treated with 20 mM ethylenediaminetetraacetic acid (EDTA, pH 7.5) at 20 °C for 18 h. The cartilage pieces were cleaned with deionized water, and EDTA-free pieces were stored at −20 °C until use. Sternal cartilage was soaked in 50 mM acetic acid for 30 days and then homogenized at 10,000 rpm on a homogenizer (X40/38 E3, Ystral, Ballrechten-Dottingen, Germany) for 30 min on ice at 4 °C. The acid-soaked cartilage was extracted with 1% pepsin (Sigma-Aldrich, St. Louis, MO, USA) and 1% protease M (Amano M, Amano Enzyme Co., Ltd., Tokyo, Japan) or without protease at 15 °C for 72 h while stirring. The mixture was filtered to remove undissolved particles and then lyophilized. The pepsin-soluble, Amano M–soluble, and unproteolyzed C-II contained 74.5, 86.9, and 62.5 mg/g hydroxyproline for 596, 695.2, and 500 mg/g collagen. The presence of epitopes in undenatured C-II was measured using the Chondrex Type II Collagen Detection Kit in accordance with the manufacturer’s protocol (Chondrex, Redmond, WA, USA). The results revealed that the percentage of epitopes in pepsin-soluble C-II, Amano M–soluble C-II, and unproteolyzed C-II was 68.4%, 80.5%, and 28.0%, respectively. Pepsin-soluble C-II and Amano M–soluble C-II were fortified with bovine collagen peptide (type I collagen, MW 300–8000 Da, Nippi, Fujinomiya, Shizuoka Prefecture, Japan) to achieve ratios of type I collagen to C-II of 5.25 to promote the dispersion of C-II fibrils.

### 2.2. Animal Preparation

For this study, 8-week-old male Wistar rats were procured from BioLASCO Taiwan (Taipei, Taiwan). The animal room was set to a photoperiod of 12-h light–dark cycle, with the humidity and temperature maintained at 50–55% and 24 ± 1 °C, respectively, using an air conditioning system throughout the experimental period. During the experiment, the rats weighed approximately 300 ± 10 g and were between 9 and 10 weeks old. Regardless of the experimental mode, all surgical procedures and feeding of the rats were completed after they were anesthetized with 2.5% isoflurane (catalog no. 08547, Panion & BF Biotech, Taoyuan, Taiwan). The handling and experimental use of animals conformed to the Guiding Principles for the Care and Use of Animals of the American Physiological Society, and the study protocol was approved by the Institutional Animal Care and Use Committee of National Sun Yat-sen University (approval number: 10725).

### 2.3. Establishment of Animal Models

Animal models of anterior cruciate ligament transection (ACLT)–induced OA were established using the methods proposed by Stoop et al. (2001) and Yang et al. (2014) [21,22]. In this study, first, the baseline activity of the rats was measured; the rats were then anesthetized, and their right knee joint was shaved. Methanol was used for sterilization, and the knee joint capsule of each rat was opened through a medial parapatellar incision. The patella was dislocated laterally, and the knee joint was fully flexed to expose the cruciate ligament. An incision was made anterior to the medial collateral ligament, and the anterior drawer test was implemented to verify the adequacy of the cut before surgical suturing. In the sham group, the knee joint capsule was opened, but no incision was made anterior to the medial collateral ligament [21]. After surgery, the rats were injected with cefazolin (20 mg/kg) to prevent wound infection and were returned to their cages to recover for eight weeks. Weekly testing was implemented to analyze differences between the experimental and control groups regarding pain, inflammation, and knee swelling. C-II treatment was administered after significant pain, inflammation, and swelling differences were observed between the experimental and control groups.

### 2.4. Experimental Groups

The rats were randomly assigned to six experimental groups.

Control: naïve rats (*n* = 8).

Sham: ACL was exposed but not transected (*n* = 8).

ACLT: ACLT of the right knee (*n* = 8).

Group A (ACLT + pepsin-soluble C-II fortified with type I collagen): Rats undergoing ACLT were orally administered 0.25 mg/kg/day collagen (containing 0.04 mg/kg/day C-II and 0.21 mg/kg/day type I collagen), which was dissolved in 1 mL of ultrapure water, once daily for 17 consecutive weeks beginning eight weeks after ACLT (*n* = 8).

Group B (ACLT + Amano M–soluble C-II fortified with type I collagen): Rats undergoing ACLT were orally administered 0.25 mg/kg/day collagen (containing 0.04 mg/kg/day C-II and 0.21 mg/kg/day type I collagen), which was dissolved in 1 mL of ultrapure water, once daily for 17 consecutive weeks beginning eight weeks after ACLT (*n* = 8).

Group C (ACLT + high-dose Amano M–soluble C-II fortified with type I collagen): Rats undergoing ACLT were orally administered 0.75 mg/kg/day collagen (containing 0.12 mg/kg/day C-II and 0.63 mg/kg/day type I collagen), which was dissolved in 1 mL of ultrapure water, once daily for 17 consecutive weeks beginning eight weeks after ACLT (*n* = 8).

Group D (ACLT + unproteolyzed C-II): Rats undergoing ACLT were orally administered 0.24 mg/kg/day unproteolyzed collagen (containing 0.12 mg/kg/day C-II), which was homogeneously suspended in 1 mL of ultrapure water, once daily for 17 consecutive weeks beginning eight weeks after ACLT (*n* = 8).

### 2.5. Analysis of Pain Behavior of Rats

#### 2.5.1. Weight-Bearing Distribution in Hind Legs

The difference in weight-bearing distribution between the hind leg with ACLT-induced joint degeneration and the contralateral leg is a pain indicator in OA [23]. A dual-channel weight averager (Singa Technology, Taoyuan, Taiwan) was used to assess the weight-bearing distribution of the rats’ hind legs. First, each rat was placed on the sloped channel with its hind legs resting on two weight-averaging platform pads. The researchers ensured the rat was stationary and its posture stable. The measurement button on the instrument was pressed to record the distribution of the animal’s body weight on each paw for 3 s. The results are presented as the difference between the amount of weight placed on the uninjured paw (i.e., left paw) and the amount placed on the injured paw (i.e., right paw) [24]. The baseline activity levels of the rats were measured weekly post-surgery from weeks 1 to 24.

#### 2.5.2. Secondary Mechanical Allodynia

Allodynia is pain caused by normally non-noxious stimuli. This study assessed mechanical allodynia by employing von Frey hair monofilaments (27 inches; Touch-Test Sensory Evaluators, NC12775, NorthCoast Medical, Morgan Hill, CA, USA) with stiffness ranging between 2.0 and 28.8 g in combination with Dixon’s up-and-down method. The hair monofilaments were applied to the plantar surface of each hind paw in ascending order of stiffness, and whether the rats exhibited a reflex response was observed. If no response occurred, von Frey hairs of increasing stiffness levels were applied until the rat exhibited a reflex response, and the stiffness level was recorded [24]. Mechanical allodynia testing was implemented to measure reflex responses in the rats at baseline and weekly from weeks 1 to 24.

#### 2.5.3. Knee Swelling

Changes in knee swelling can reflect the severity of inflammation in the knee joint. To assess the severity of inflammation in the knee joint, the circumference of each rat’s knee joint was measured before surgery (baseline) and weekly from weeks 1 to 24 post-surgery. After the rats were anesthetized with 2.5% isoflurane, a Vernier scale (calipers, AA847R, Aesculap, Tuttlingen, Germany) was used to measure the widths of the rats’ knee joints, and the changes in width were recorded [22].

### 2.6. Sample Collection and Fixation

After the animal behavior experiments, the rats were euthanized for further analysis. Each rat was anesthetized with 2.5% isoflurane, and the chest was opened using surgical instruments. The rats were injected with 600 mL of 4% phosphate-buffered saline (PBS; 137 mM NaCl, 2.68 mM KCl, 10 mM Na_2_HPO_4_, and 1.76 mM KH_2_PO_4_; pH = 7.2; storage temperature = 4 °C) and heparin (0.2 U/mL) by using a perfusion needle inserted through the ventricle and into the left aorta with a pump. An incision was made in the right atrium to enable blood outflow, achieving full-body blood replacement. Subsequently, 4% paraformaldehyde (at a storage temperature of 4 °C) was injected in situ for tissue fixation. Finally, the knee joint tissue was extracted using surgical instruments. The extracted tissue samples were maintained in 10% neutral buffered formalin at 4 °C for 3 to 4 days, during which time the solution was replaced every two days to maintain its fixative effect.

### 2.7. Micro-Computed Tomography Scanning Analysis

This study adapted the research methods of Bagi et al. [25] with modification. Before completing the bone samples’ microcomputed tomography (micro-CT) scans (SkySacn 1076, Bruker, Antwerp, Belgium), the Taiwan Mouse Clinic was commissioned to reconstruct 3D animal skeleton models and implement data analysis of the reconstructed images. Micro-CT scanning, which was implemented with a resolution of 35 μm (isotropic voxel size: 35 μm), was used to analyze the 3D microstructure parameters of statistical significance, including trabecular separation, trabecular number, bone mineral density (BMD), tissue volume, bone volume, and the ratio of bone volume to tissue volume.

### 2.8. Sample Embedding

The fixated tissues were immersed in decalcification solution (100-g EDTA disodium salt dehydrate/1000-mL PBS) at room temperature for 2 to 3 weeks, during which the solution was replaced every two days to maintain its effectiveness for removing calcium deposits. The fixated and decalcified tissue samples were placed into tissue cassettes and processed using an automatic tissue dehydration machine (Tissue-Tek, Sakura Finetek Japan, Tokyo, Japan). The tissue samples were treated (in the following order) with 35% alcohol, 75% alcohol, 85% alcohol, 85% alcohol, 95% alcohol, 95% alcohol, 100% alcohol, l00% alcohol, 90% xyline/alcohol, 100% xyline, soft paraffin, and hard paraffin. The tissue dehydration and paraffin embedding process lasted 18 h. Finally, an embedding center (CSA Embedding Center EC780-1; EC780-2, Pomona, CA, USA) was used to embed the tissues into tissue blocks. A paraffin slicing machine (Microm, HM340E, Kentwood, MI, USA) was then used to slice the tissue blocks into 1-µm slices, which were stained.

### 2.9. Histological Staining Analysis

To evaluate the cartilage tissue samples, the grading system for safranin-O/fast green staining proposed by the Osteoarthritis Research Society International (OARSI) was employed. Semi-quantitative analysis was implemented using the OARSI grading system (i.e., six histological grades) while considering the rats’ OA recovery stages (i.e., four histological stages). The total scores of the samples were calculated as the product of the historical grade and the histological stage and ranged from 1 (normal articular cartilage) to 24 (no repair) [26]. Synovial tissue was analyzed using the tissue assessment method proposed by Krenn et al. (2006), which involves hematoxylin and eosin (H&E) staining [27]. The grading criteria for the synovial lining layer were the presence of subsynovial fibroblasts (0–3 points), synovial hyperplasia (0–3 points), and red blood cell infiltration (0–3 points). The grading criteria for subsynovial tissue were the formation of granulation tissue (0–3 points), angioplasia (0–3 points), and red blood cell infiltration (0–3 points). The severity of inflammation in synovial tissue was divided into three stages; mild, moderate, and severe inflammation were indicated by scores of 0–6, 7–12, and 13–18, respectively. A higher total score represents greater damage to the knee joint. The glass slides of the stained samples were examined using an optical microscope (DM 6000, Leica, Wetzlar, Germany) equipped with a digital microscope camera (SPOT Idea, Diagnostic Instruments, Sterling Heights, MI, USA).

### 2.10. Immunohistochemical Staining Analysis

In the immunohistochemical staining experiment, the paraffin slides were first immersed in xylene solution for 20 min and then in ethanol with a concentration ranging from 50% to 100% for 30 s. Finally, the paraffin slides were subjected to proteinase K (20 mM; Sigma-Aldrich, St. Louis, MO, USA) digestion for 20 min. Each slide was rinsed twice with tris-buffered saline with Tween (TTBS), encircled with a Dako pen (Dako, S2002), and immersed in hydrogen peroxide for 8 min. The slides were again rinsed twice with TTBS, immersed in the ABC Kit blocking buffer (Vectastain ABC Kit, Vector Laboratories, Burlingame, CA, USA), and oscillated at room temperature for 30 min. The samples were subsequently incubated with a primary antibody specific to the target protein and oscillated overnight at 4 °C. The sections were then incubated with specific primary antibodies, namely anti-C-II antibodies (1:200; catalog no. 234187; Calbiochem, San Diego, CA, USA), anti-matrix metallopeptidase 13 (MMP13) antibodies, (1:100; catalog no. ab39012; Abcam, Cambridge, UK), anti-interleukin-1β antibodies (IL-1β 1:200; catalog no. ab9722; Abcam), and anti-tumor necrosis factor-α (TNF-α antibodies (1:150; catalog no. ab6671; Abcam). The slides were then rinsed twice with TTBS, immersed in the aforementioned blocking buffer, and oscillated at room temperature for 30 min. Next, the slides were incubated with a secondary antibody (Vector Laboratories, Burlingame, CA, USA) in the ABC Kit, oscillated at room temperature for 80 min and in fresh TTBS solution three times for 10 min each. They were subsequently immersed in the aforementioned blocking buffer, oscillated at room temperature for 30 min, and washed once with fresh TTBS. The samples were then incubated with DAB staining solution (DAB Peroxidase Substrate Kit, Vector Laboratories) for 5 min, rinsed twice, soaked in Mayer’s hematoxylin for 90 s, rinsed for 5 min, and soaked sequentially in ethanol solutions with concentrations of 50%, 70%, 95%, and 100% for 20 s each. Finally, the slides were immersed in xylene solution for 1 min before being sealed with a coverslip. The slides were imaged under an optical microscope (DM 6000B, Leica, Wetzlar, Germany) equipped with the aforementioned digital microscope camera (SPOT Idea 5.0 Mp 635 Color Digital Camera, Diagnostic Instruments, Sterling Heights, MI, USA).

### 2.11. Data Analysis

The experimental data are presented as the mean ± standard error of the mean. For statistical analyses, we calculated the variation between groups by a one-way analysis of variance (ANOVA), examined by a post hoc Tukey test. We defined the statistical significance as *p* < 0.05. Statistical analyses were performed using SigmaPlot Version 12.0 (Systat Software, Inc., San Jose, CA, USA).

## 3. Results

### 3.1. Effect of (Oral Protease-Soluble Chicken C-II) PSCC-II on ACLT-Induced Weight-Bearing Deficits

A dual-channel weight averager was employed to determine the weight-bearing distribution of the rats’ hind legs. Figure 1A presents the difference in ACLT-induced weight-bearing deficits between the rats’ hind legs. The results revealed a significant difference in the weight-bearing distribution of the hind legs between the ACLT and sham groups from weeks 1 to 24 post-surgery. Among the rat models of ACLT-induced OA, the differences in the weight-bearing distribution of the hind legs in groups A, B, C, and D were significantly lower than that in the ACLT group from weeks 12–24, 12–24, 11–24, and 11–24 post surgery, respectively. In summary, significant improvements in the ACLT-induced weight-bearing deficit in the hind legs were detected in treatment groups A, B, C, and D.

### 3.2. Effect of PSCC-II on ACLT-Induced Mechanical Allodynia

Figure 1B presents the effects of PSCC-II on ACLT-induced mechanical allodynia. The results revealed that the paw withdrawal threshold (g) of the ACLT group was significantly lower than that of the sham group post-surgery. Among the groups with ACLT-induced OA, the paw withdrawal thresholds (g) of groups A, B, C, and D were significantly higher than that of the ACLT group from weeks 13–24 post-surgery. In summary, treatment groups A, B, C, and D had reduced considerably mechanical allodynia caused by ACLT-induced OA. Notably, the treatment effect in group C was higher than in the other groups but did not attain statistical significance.

### 3.3. Effect of PSCC-II on ACLT-Induced Knee Joint Swelling

The knee joint width of the hind legs was measured using a Vernier scale. The results, presented in Figure 1C, revealed that the difference in the knee joint width of the ACLT group was significantly greater than that of the sham group from weeks 2 to 24 post-surgery. Among the groups with ACLT-induced OA, the differences in the knee joint width of groups A, B, C, and D were significantly smaller than that of the ACLT group from weeks 12–24, 12–24, 12–22, and 13–24 postsurgery, respectively. In summary, treatment groups A, B, C, and D had significantly reduced swelling of the knee joint caused by ACLT-induced OA.

### 3.4. Effect of PSCC-II on Body Weight

Figure 1D presents the effects of PSCC-II on body weight. The weights of all groups increased over time. The results revealed no significant differences in body weight between the ACLT and sham groups from weeks 0 to 24 post-surgery. No significant differences were found in body weight between the treatment groups (A, B, C, and D) and the ACLT group from weeks 9 to 24 post-surgery. In summary, no significant differences in body weight were observed between groups A, B, C, and D.

### 3.5. Micro-CT Analysis of Knee Bone Structure in Rats Undergoing ACLT after PSCC-II Treatment

Figure 2A presents 2D micro-CT images of knee joints displaying the tibial cancellous bone. The images reveal greater tibial cancellous bone loss in the ACLT group than in the sham group. The restoration effects in groups A and D were not greater than those in the ACLT group; however, the restoration effects in groups B and C were markedly greater than those in the ACLT group. Micro-CT scanning was employed to quantitatively analyze bone volume (Figure 2B), trabecular separation (Figure 2C), trabecular number (Figure 2D), and BMD (Figure 2E) in the tibial cancellous bone. These parameters were evaluated using reconstructed 3D images of the tibial metaphysis. The results indicated that the ACLT group had significantly lower cancellous bone volume, lower trabecular number, and increased trabecular separation than the sham group. Treatment groups A, B, and C had significantly greater restoration effects than the ACLT group with respect to cancellous bone volume, trabecular number, and trabecular separation; however, the results did not indicate a significant restoration effect in group D. Figure 3 presents the reconstructed 3D images of the subchondral bone (SB) for the control, sham ACLT, and ACLT plus treatment groups. Table 1 presents the quantitative analysis of tissue volume, bone volume, the ratio of bone volume to tissue volume, and BMD of the reconstructed 3D images of the SB. Analysis of the reconstructed 3D images revealed no significant differences between groups. In summary, PSCC-II treatment administered to groups A, B, and C effectively mitigated cancellous bone loss caused by ACLT-induced OA.

### 3.6. Effect of PSCC-II on Synovial Tissue Inflammation in Knee Joints Subjected to ACLT

Figure 4A–H present the pathological H&E staining results. Synovial hyperplasia and red blood cell infiltration were significantly greater in the ACLT group than in the sham group. The rats in the ACLT group had severe knee joint inflammation. The rats in group A had mild synovial inflammation, and synovial hyperplasia and neutrophil infiltration were significantly less severe in group A than in the ACLT group. Similar results were observed in groups B, C, and D. PSCC-II treatment effectively alleviated synovial hyperplasia and red blood cell infiltration in knee joints subjected to ACLT in groups A, B, C, and D.

### 3.7. Effect of PSCC-II on Articular Cartilage Degradation in Knee Joints Subjected to ACLT

Figure 4I–O presents the safranin-O/fast green staining of knee articular cartilage. Cartilage histopathology was further assessed using the OARSI histological scoring system (Figure 4P). The results revealed that the ACLT group had greater damage to the surface of cartilage tissue, greater loss of chondrocytes, and lower staining intensity of cartilage than the sham group. These pathological changes in articular cartilage were reduced in the ACLT plus treatment groups (i.e., A, B, C, and D). The OARSI score was significantly higher in the ACLT group (9.75 ± 0.94) than in the sham group (0.00 ± 0.00). Groups A, B, C, and D had significantly lower OARSI scores than the ACLT group (2.57 ± 0.53, 0.63 ± 0.26, 0.43 ± 0.20, 3.83 ± 0.65, respectively). Therefore, PSCC-II treatment effectively alleviated articular cartilage degradation in the rats with ACLT-induced OA in groups A, B, C, and D. PSCC-II treatment almost completely counteracted ACLT-induced articular cartilage degradation in groups C.

### 3.8. Effect of PSCC-II on Expression of C-II and MMP-13 in Articular Cartilage in Knee Joints Subjected to ACLT

Figure 5A–H presents the effect of PSCC-II on C-II immunoreactivity in knee articular cartilage. C-II immunoreactivity was lower in the ACLT group than in the sham group. C-II expression was increased in the ACLT plus PSCC-II groups (i.e., groups A, B, and C) and group D compared with that in the ACLT group. Quantitative analysis of C-II expression in articular cartilage revealed significantly decreased immunoreactivity in the ACLT group (0.49 ± 0.08 folds) compared with the sham group (0.97 ± 0.10). The results also revealed significantly increased immunoreactivity in groups A, B, C, and D (1.74 ± 0.11, 1.98 ± 0.11, 2.05 ± 0.14, 1.44 ± 0.14 folds, respectively) compared with the ACLT group. Moreover, all treatment groups exhibited significantly increased immunoreactivity compared with the sham group. In summary, the decreased ACLT-induced C-II expression in articular cartilage was mitigated significantly in groups A, B, C, and D. Figure 5I–P presents the effects of PSCC-II on MMP-13 immunoreactivity in knee articular cartilage. MMP-13 immunoreactivity was higher in the ACLT group than in the sham group. All treatment groups exhibited decreased *MMP-13* expression compared with the ACLT group. Quantitative analysis of *MMP-13* expression in articular cartilage revealed significantly increased immunoreactivity in the ACLT group (48.76% ± 2.59%) compared with the sham group (8.55% ± 1.44%). Immunoreactivity was significantly lower in groups A, B, C, and D (25.67% ± 3.91%, 28.34% ± 1.28%, 20.68% ± 1.42%, 22.93% ± 2.90%, respectively) compared with in the ACLT group. In summary, PSCC-II treatment in groups A, B, and C and the treatment in group D effectively mitigated the decreased *MMP-13* expression caused by ACLT.

### 3.9. Effect of PSCC-II on Expression of TNF-α and IL-1β in Chondrocytes in Articular Cartilage after ACLT

Figure 6A–P present the effects of PSCC-II on the immunoreactivity of TNF-α and IL-1β in chondrocytes within articular cartilage. The immunoreactivity of TNF-α and IL-1β was higher in the ACLT group than in the control and sham groups. The results revealed the decreased expression of TNF-α and IL-1β in groups A, B, C, and D compared with that in the ACLT group. Quantitative analysis of TNF-α and IL-1β expression revealed significantly increased immunoreactivity in the ACLT group compared with that in the sham group. Immunoreactivity was significantly decreased in the treatment groups compared with the ACLT group. In summary, PSCC-II treatment in groups A, B, and C and the treatment in group D effectively inhibited the reduction in TNF-α and IL-1β expression caused by ACLT. Notably, the results indicated no significant difference in the immunoreactivity of TNF-α and IL-1β between the sham group and group C.

## 4. Discussion

This study investigated the potential protective effects of orally administered PSCC-II against nociception, histopathological cartilage degradation, and the expression of proinflammatory cytokines in rats with ACLT-induced OA. Additionally, we employed micro-CT to assess the SB microstructure in the tibial knee region. The results revealed that the oral administration of PSCC-II and undenatured C-II effectively reduced nociception sensitivity in rats with ACLT-induced OA. These supplements mitigated weight-bearing deficits, mechanical allodynia, knee joint swelling, cartilage degradation, synovitis, and the expression of ECM degradative enzymes and proinflammatory cytokines in knee articular cartilage. Notably, PSCC-II prevented microstructural changes induced by ACLT in the proximal tibial metaphysis, such as reduced bone volume and trabecular number and increased trabecular separation. However, undenatured C-II did not prevent these changes.

Rats with ACLT-induced OA are a widely used model for studying OA. ACLT in rats leads to knee joint inflammation, neutrophil infiltration, SB remodeling, and osteophyte formation. All these factors can exacerbate the occurrence of OA [28]. Cartilage degradation is simulated in the ACLT-induced OA model, as ACL injury causes long-term joint instability. The progression of this OA model is slow; however, the study observed molecular changes in cartilage tissue, synovitis, and SB sclerosis similar to those in human OA, which may be conducive to drug research [29]. Another study suggested that the experimental model of ACLT alone may be more suitable for evaluating the potential of drugs to alleviate OA [30]. The results of the present study revealed that treatment with oral PSCC-II (groups A, B, and C) and undenatured C-II (group D) significantly inhibited ALCT-induced weight-bearing deficits, mechanical allodynia, and knee joint swelling in rats with OA (Figure 1A–C). PSCC-II and undenatured C-II did not adversely affect body weight (Figure 1D).

Studies have reported that protein glycosylated side chains may play a crucial role in the overexcitation of T cells and contribute to the immune response [11,31]. Studies employing C-II-induced arthritis (CIA) animal model have contended that C-II is a cartilage-specific autoantigen and that its glycosylated side chains can promote T cell overactivation [32,33]. CIA is a common animal model used to study RA. Studies have verified that naïve glycosylated C-II is more likely to induce arthritis and cause changes in articular cartilage and SB than non-glycosylated C-II. Removing or modifying carbohydrates from C-II could reduce the incidence, onset time, and severity of CIA in animals [11,12]. Clinical studies have revealed that 3–27% of patients with RA have serum antibodies against C-II; their results also revealed a positive correlation between T cell responses to C-II and disease severity [15,34]. Undenatured C-II is an untreated bundle of native C-II protein fibers obtained by purifying animal cartilage. The procollagen form of C-II contains three identical alpha chains; each has extensions called non-collagenous telopeptides and retains its glycoprotein side chain structure [35]. One study revealed that collagen-IX and C-II exhibit polysaccharide cross-linking within cartilage tissue, forming a robust fibril structure [16]. The use of proteases can increase extraction efficiency because they target and degrade the side chains of polysaccharides on collagen molecules, facilitating the release of C-II from the fibril structure [17]. However, telopeptides of procollagen are insoluble and may contribute to the immune response [10]. Modification of glycosylated side chains of C-II can also alleviate the excessive activation of the immune system [8]. Therefore, the proteolytic action of enzymes in C-II may involve the modification of glycosylated side chains and the removal of telopeptides, thereby enhancing solubility and reducing adverse immune reactions. In our unpublished experimental results, the molecular weight of undenatured C-II was approximately 300 kDa, whereas that of PSCC-II was approximately 280–300 kDa. We speculate that the telopeptides and glycosylated structures on the C-II fiber bundles of PSCC-II obtained from enzymatic proteolysis may change, potentially leading to reduced autoimmunity. This speculation merits additional investigation.

Studies have widely leveraged micro-CT’s high-resolution and 3D reconstruction capabilities for evaluating animal and human OA histopathology [36]. Changes in the SB structure in joint regions, a histopathological characteristic of OA, can contribute to the load-bearing pressure experienced by the body after cartilage degeneration [37]. In OA, the SB undergoes structural changes, such as decreased bone volume and trabecular number and increased trabecular separation [38]. Structural changes in the SB interact synergistically with articular cartilage degradation; thus, alterations in the SB structure can lead to secondary cartilage injury and degradation. Cartilage damage or loss can alter the load-bearing capacity of the underlying SB [38]. The trabecular bone, a component of SB, has a layered, spongy, and porous structure and crucially contributes to the load-bearing capacity of the body; its mechanical supporting properties are influenced by the mineral content and type I collagen [39]. In 1998, Fazzalari and Parkinson found a significant decrease in trabecular number and increased trabecular separation of the SB in patients with OA [40]. In 2008, Kadri et al. observed a significant decrease in bone volume and an increase in trabecular separation in mice with OA [41]. Bagi et al. employed a rat model in which a partial medial meniscectomy tear was surgically induced to trigger OA development; that study also revealed reduced trabecular bone volume in the SB [25]. Our experimental results align with the findings of these relevant studies. The ACLT-induced OA group exhibited a significant decrease in bone volume and trabecular number (Figure 2B,D) and a significant increase in trabecular separation (Figure 2C). Oral administration of PSCC-II (i.e., groups A, B, and C) mitigated these structural changes in the SB, and the results of group D were superior to those of groups A and B. However, undenatured C-II (group D) treatment did not significantly affect the changes in bone volume, trabecular number, and trabecular separation caused by ACLT.

Type I collagen comprises 90% of the total protein in the SB, and it is influenced by factors such as mechanical loading, pathological insults, and bone types [42,43,44]. Studies have revealed that patients with osteogenesis imperfecta (OI) primarily exhibit mutations in the collagen type 1 alpha 1 chain (*COL1A1*) gene or the collagen type 1 alpha 2 chain (*COL1A2*) gene, which are associated with the lower trabecular number, trabecular thickness, bone mass, and connectivity in the SB. These findings indicate a significant positive correlation between structural changes in the SB and the expression of type I collagen [45]. Wu et al. (2020) asserted that type I collagen is unevenly distributed and is significantly reduced in the SB of patients with OA. This decrease in type I collagen is accompanied by a decrease in trabecular bone volume and an increase in trabecular separation [46]. In the present study, groups A, B, and C received PSCC-II treatment. During the preparation process of PSCC-II, in addition to proteolytic C-II, type I collagen was added to dissolve PSCC-II. The type I collagen dosage in group C was 0.63 mg/kg/day, higher than in groups A and B (0.21 mg/kg/day). The results revealed that the decrease in bone volume caused by ACLT was more effectively mitigated in group C than in groups B and C (Figure 2B). Therefore, in addition to the proteolytic C-II employed to reduce the risk of adverse immune reactions, the additional type I collagen may counteract the ACLT-induced structural changes in the SB. We predict that PSCC-II can mitigate changes in the SB in patients with OA and can be applied to prevent OI, which should be investigated in future studies.

As early as 1982, Goldenberg et al. revealed an association between OA and synovitis of the knee joint [47]. Cartilage degradation is promoted by synovitis-induced proinflammatory cytokines, such as TNF-α, IL-1β and IL-6, and extracellular matrix-degrading enzymes (i.e., MMPs), resulting in OA progression [48]. In a 2020 retrospective study, Hasan et al. contended that undenatured C-II alleviates synovitis and cartilage degradation in humans, horses, dogs, and rodents with OA [8]. In the present study, the rats with ACLT-induced OA exhibited ACLT-induced synovitis characterized by synovial tissue thickening and increased infiltration of blood cells. The rats also showed cartilage degradation, with losses in proteoglycans and C-II. All these phenomena were attenuated by orally administered PSCC-II (groups A, B, and C) and undenatured C-II (group D; Figure 4). These treatments also significantly inhibited the expression of proinflammatory IL-1β and TNF-α proteins and the ECM-degrading protein MMP13 in cartilage tissue after ACLT (Figure 5P and Figure 6H,P). The pro-inflammatory cytokines IL-1β and TNF-α inhibit the synthesis of major extracellular structural proteins, such as C-II and aggrecan. They simultaneously stimulate chondrocytes to produce MMPs (e.g., MMP-1, MMP-3, and MMP-13), disintegrin and metalloproteinase with thrombospondin motifs, which lead to ECM degradation. This results in cartilage degradation and exacerbates OA progression [49,50,51].

In summary, this study demonstrated that the oral administration of PSCC-II (groups A, B, and C) and undenatured C-II (group D) enhanced the quantity of C-II in the ECM by suppressing the expression of proinflammatory cytokines and MMP-13. In a 2009 study, cartilage and synovial tissue from patients with OA were cocultured; that study revealed that the administration of type I collagen upregulated the expression of proteoglycans, C-II, and the anti-inflammatory cytokine IL-10, and downregulated the expression of the proinflammatory cytokines IL-1β and TNF-α [52]. Dar et al. (2017) verified that the daily oral administration of hydrolyzed type I collagen mitigated the degradation of cartilage tissue and the expression of MMP13 and TNF-α in mice with meniscal plus ligamentous injury-induced OA, and hydrolyzed type I collagen also inhibited synovial thickening [53]. In the present study, the PSCC-II groups (groups A, B, and C) and the undenatured C-II group exhibited improved ACLT-induced OA symptoms, but the effect was greater in the PSCC-II groups. We suggest that PSCC-II exerts chondroprotective effects similar to those of undenatured C-II. It may also counteract cartilage damage and ECM degradation because it contains additional type I collagen.

Currently, three configurations of C-II are used for nutraceutical applications to prevent OA: natural insoluble undenatured C-II, protease-soluble undenatured C-II, and hydrolyzed C-II. Natural insoluble undenatured C-II can be derived from chicken [25], porcine [54], and salmon nasal cartilages [55], and it is used as a nutraceutical ingredient under names such as UC-II (Lonza, Collavant of Bioiberica, S.A.U., Esplugues de Llobregat, Barcelona) and SCP-II (Guzen Development, Walnut Creek, CA, USA). Protease (pepsin)-soluble undenatured C-II, such as EXT-II (Ryusendo Co., Ltd., Tokyo, Japan) [55], is extracted from chicken sternal cartilage. The C-II hydrolysate is primarily obtained from insoluble undenatured collagen fibrils from chicken sternal cartilage, such as in BioCell Collagen (Biocell Technology, Irvine, CA, USA) [56]. In PSCC-II, only pepsin-soluble undenatured C-II extracted from chicken sternal cartilage is available on the nutraceutical market. The use of marine pepsin-soluble undenatured C-II sourced from the skulls and cartilage of Nile tilapia and sturgeon is in the research stage [57,58]. Epidemiological studies have revealed that OA is a complex disease characterized by chronic joint pain, inflammation, synovitis, bone remodeling, and cartilage loss [59,60]. Orally administered undenatured C-II can alleviate some of the aforementioned symptoms, but fewer studies have investigated the changes in the SB caused by OA. In addition, glycosylated side chains are retained in undenatured C-II during the preparation process, which may induce an adverse immune response. PSCC-II is obtained through the enzymatic degradation of chicken sternal cartilage. In this process, pepsine or protease M and type I collagen aid in the homogenization of C-II in the final product. PSCC-II plus type I collagen may reduce adverse immune responses and mitigate structural changes in the SB in OA. C-II must be processed under acidic conditions before extraction from animal cartilage. In the present study, pepsin and protease M maintained their protease activity in acidic environments. However, pepsin, an enzyme derived from mammalian sources, is associated with the risk of transmitting certain diseases, such as transmissible spongiform encephalopathies or bovine spongiform encephalopathies. Therefore, pepsin use as a food additive has not been approved by the European Union or food safety authorities in other countries. By contrast, protease M, which is derived from the fungus *Aspergillus oryzae*, is regarded as being safer than pepsin. It has been approved as a food additive in certain countries, such as Japan and Taiwan. In the future, we propose that protease M-processed PSCC-II can be administered to patients with OA as a nutritional supplement to prevent fractures and reduce the incidence of osteoporosis.

## 5. Conclusions

Our results conclude that PSCC-II retains the protective effects of traditional undenatured C-II and provides superior benefits for OA management. These benefits encompass pain relief, anti-inflammatory effects, and the protection of cartilage and cancellous bone.

## Figures and Tables

**Figure 1 nutrients-15-03589-f001:**
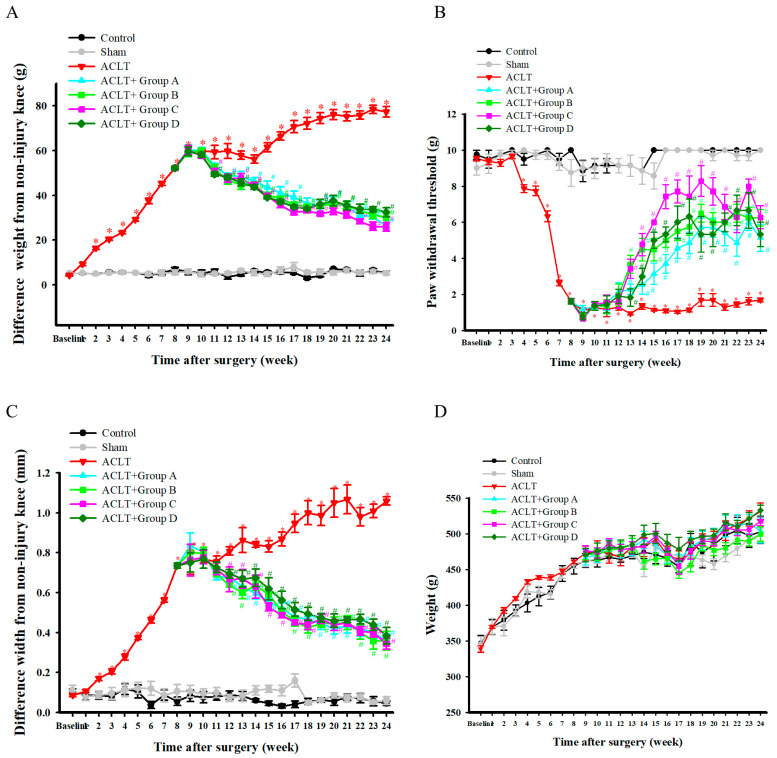
Effects of PSCC-II on ACLT-Induced OA-Related Phenomena. Effects of PSCC-II on ACLT-induced weight-bearing deficits in hindlegs (**A**), mechanical allodynia (**B**), knee joint swelling (**C**), and body weight (**D**) over time. Data are presented as mean ± standard error of the mean. * denotes comparison with sham group (*p* < 0.05); # denotes comparison with ACLT group (*p* < 0.05).

**Figure 2 nutrients-15-03589-f002:**
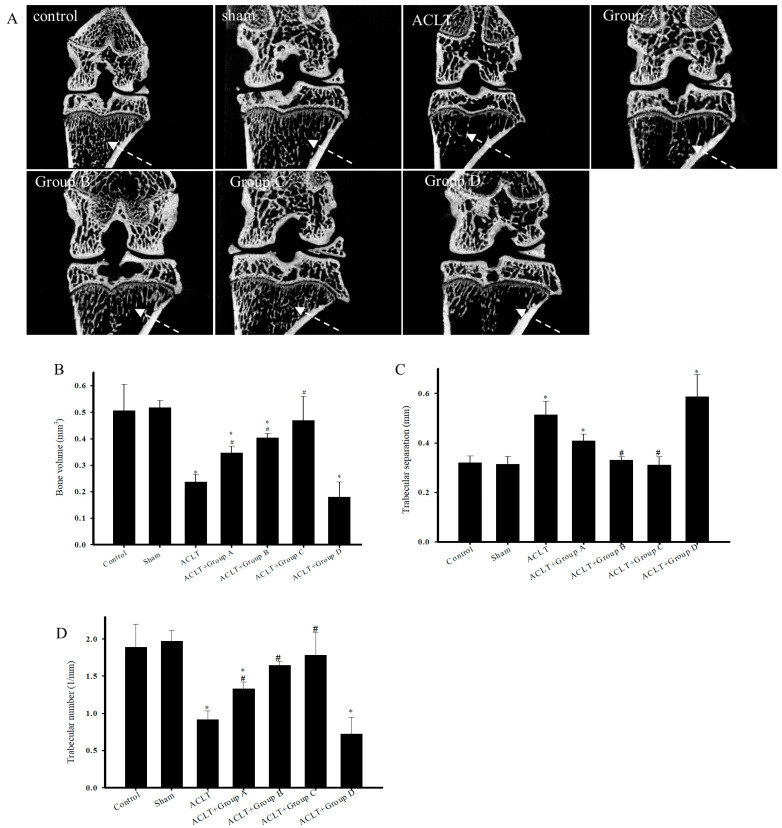
Micro-CT scans of Reconstructed Images of Cancellous Bone and Quantitative Analysis. Micro-computed tomography (micro-CT) was used to analyze the structure of the knee tibia in ACLT after treatment. (**A**) 2D micro-CT scans of reconstructed images of cancellous bone. Dotted arrows indicate image sections for comparing cancellous bone volume. Quantitative analysis of (**B**) bone volume (mm^3^), (**C**) trabecular separation (mm), and (**D**) trabecular number (1/mm). Data are presented as mean ± standard error of the mean. * denotes comparison with sham group (*p* < 0.05); # denotes comparison with ACLT group (*p* < 0.05).

**Figure 3 nutrients-15-03589-f003:**
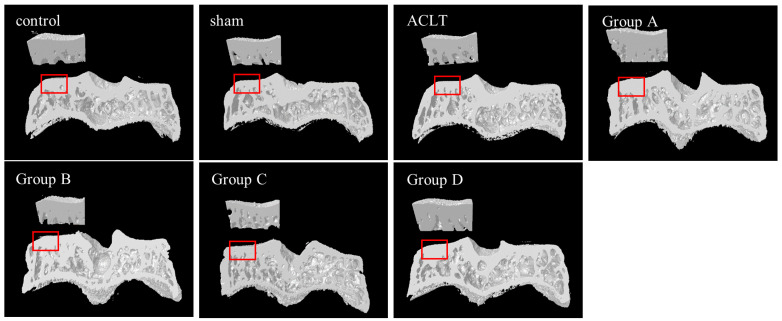
3D Micro-CT scans of Reconstructed Images of Tibial Metaphysis. Upper left sections marked with red rectangles were used for analysis. No obvious differences were observed between the sham group, ACLT group, and treatment groups A, B, C, and D.

**Figure 4 nutrients-15-03589-f004:**
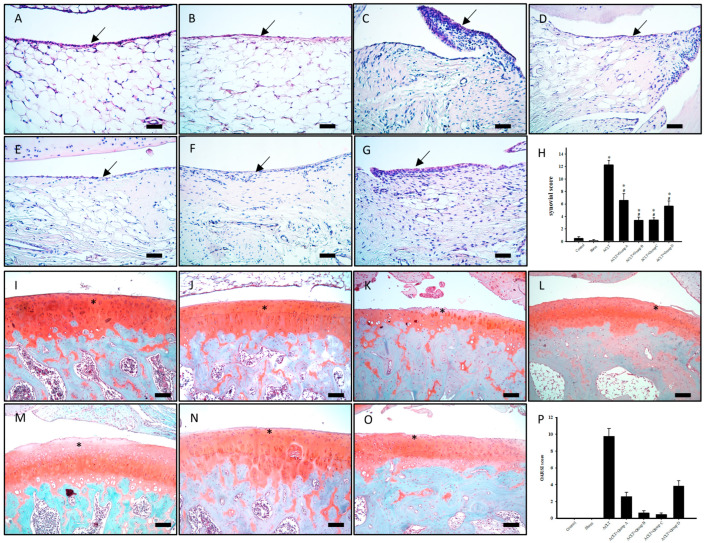
Histopathological Evaluation of Synovial Tissue and Articular Cartilage in Knee Joints Subjected to ACLT after Treatment. (**A**,**I**) indicate control group; (**B**,**J**) indicate sham group; (**C**,**K**) indicate ACLT group; (**D**,**L**) indicate group A; (**E**,**M**) indicate group B; (**F**,**N**) indicate group C; and (**G**,**O**) indicate group D. Synovial tissue samples were stained with hematoxylin and eosin stain, and cell infiltration, synovial fibrosis, angiogenesis, and synovial hypertrophy (indicated by arrows) were observed in each group. (**H**) Quantitative synovial scores for various synovial tissue samples. Cartilage degradation (indicated by an asterisk) was stained using safranin-O/fast green staining. (**P**) Quantitative histopathological changes in knee joints were evaluated using the Osteoarthritis Research Society International scoring system. Scale bar = 100 μm. Data are presented as mean ± standard error of the mean. * denotes comparison with sham group (*p* < 0.05); # denotes comparison with ACLT group (*p* < 0.05).

**Figure 5 nutrients-15-03589-f005:**
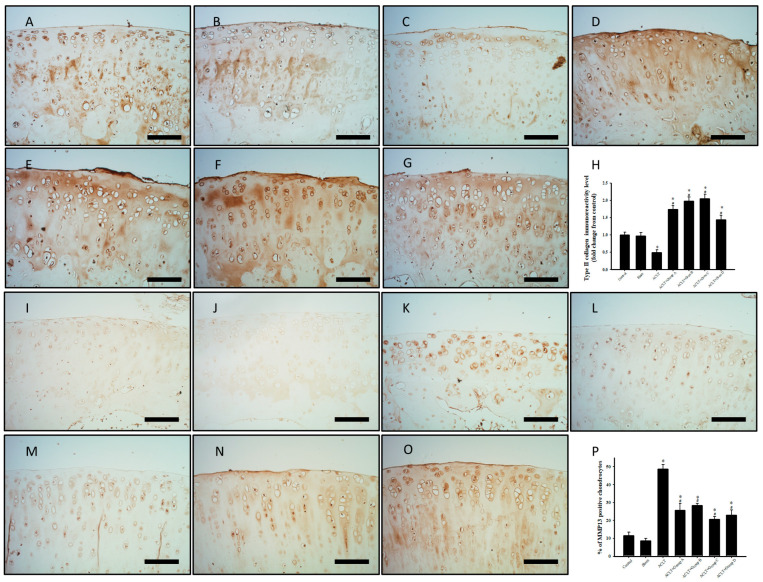
Expression of *C-II* and *MMP-13* in Cartilage Tissue with ACLT after Treatment. (**A**,**I**) indicate control group; (**B**,**J**) indicate sham group; (**C**,**K**) indicate ACLT group; (**D**,**L**) indicate group A; (**E**,**M**) indicate group B; (**F**,**N**) indicate group C; and (**G**,**O**) indicate group D. (**A**–**G**) Immunohistochemical staining of type II collagen (C-II; brown area) in knee joint sections. (**H**) Quantitative analysis of C-II expression in cartilage. (**I**–**P**) Immunohistochemical staining of MMP-13 in joint sections. Brown areas indicate immunoreactive cells. (**P**) Quantitative analysis of MMP-13–positive cells in cartilage. Scale bar =100 μm. Data are presented as mean ± standard error of the mean. * denotes comparison with sham group (*p* < 0.05); # denotes comparison with ACLT group (*p* < 0.05).

**Figure 6 nutrients-15-03589-f006:**
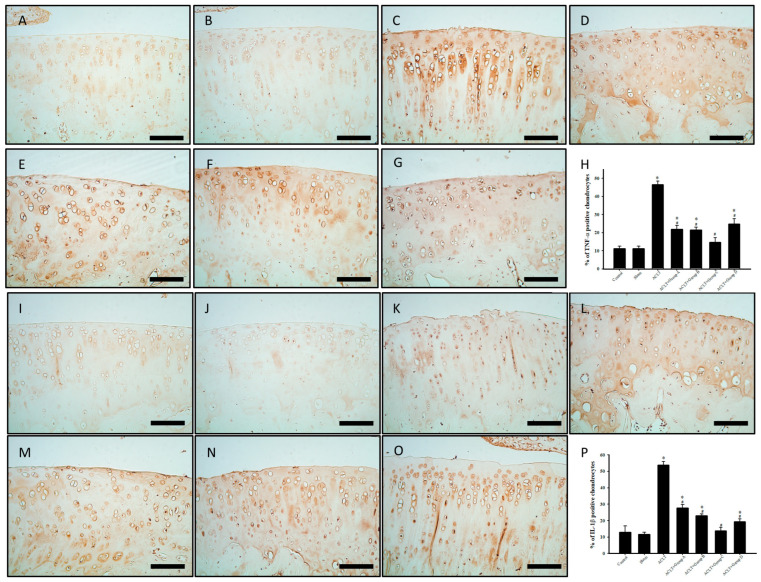
Expression of *TNF-α* and *IL-1β* in Cartilage Tissue with ACLT after Treatment. (**A**,**I**) indicate control group; (**B**,**J**) indicate sham group; (**C**,**K**) indicate ACLT group; (**D**,**L**) indicate group A; (**E**,**M**) indicate group B; (**F**,**N**) indicate group C; and (**G**,**O**) indicate group D. Brown areas indicate immunoreactive cells. (**A**–**G**) Immunohistochemical staining of TNF-α in cartilage. (**H**) Quantitative analysis of TNF-α–positive cells. (**I**–**P**) Immunohistochemical staining of IL-1β in cartilage. (**P**) Quantitative analysis of TNF-α–positive cells. Scale bar = 100 μm. Data are presented as mean ± standard error of the mean. * denotes comparison with sham group (*p* < 0.05); # denotes comparison with ACLT group (*p* < 0.05).

**Table 1 nutrients-15-03589-t001:** Analysis of 3D Micro-CT scans of Reconstructed Images of Tibial Subchondral Bone.

Parameter	Unit	Control	Sham	ACLT	ACLT + Group A	ACLT + Group B	ACLT + Group C	ACLT + Group D
Tissue vol.	mm^3^	0.492 ± 0.001	0.490 ± 0.001	0.491 ± 0.001	0.490 ± 0.000	0.490 ± 0.000	0.490 ± 0.000	0.490 ± 0.000
Bone vol.	mm^3^	0.31 ± 0.01	0.32 ± 0.02	0.31 ± 0.02	0.37 ± 0.02	0.31 ± 0.02	0.37 ± 0.02	0.32 ± 0.04
BV/TV	%	63.92 ± 2.05	66.30 ± 3.238	62.24 ± 3.06	75.02 ± 4.53	63.88 ± 3.89	75.35 ± 4.85	66.06 ± 8.09
BMD	g/cm^3^	0.726 ± 0.014	0.711 ± 0.020	0.691 ± 0.013	0.743 ± 0.013	0.684 ± 0.016	0.767 ± 0.023	0.687 ± 0.04

BV/TV, bone vol./tissue vol.; BMD, bone mineral density.

## Data Availability

Correspondence and requests for materials should be addressed to Z.-H.W. and Y.-P.H.

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
