# Peer review of "Oral Administration of Protease-Soluble Chicken Type II Collagen Ameliorates Anterior Cruciate Ligament Transection–Induced Osteoarthritis in Rats"

_nutrients, 2023, doi:10.3390/nu15163589_

Round 1

Reviewer 1 Report

ABSTRACT

Lines 25-27 – This phrase is unnecessary and should be removed. The Abstract should begin at “This study…”

Line 35 – Suggestion: Groups that received (instead of “Groups receiving”)

Lines 38-39 – The phrase is unclear. Did the authors mean that all treatments suppressed the ACTL-induced effects (downregulation of C-II expression, and upregulation of MMP13, TNF-a and IL-1b expression)? Please, rewrite.

Line 40 – Why PSCC-II treatment “may provide superior benefits for OA management”? Either explain here or remove the phrase.

INTRODUCTION

Lines 68 and 75 – “naïve” collagen – did the authors mean “native” collagen?

Line 81 – What is “oral tolerance properties”?

RESULTS

Lines 276-277 – Please, remove this phrase.

Page 8, Figure 1 – “B” and “C” – it seems that the legend does not corresponds to the Figures. It seems that Figure 1-B is the mechanical allodynia, while Figure 1-C is swelling. Please, check.

CONCLUSIONS

Line 601 – There is only one Conclusion. So, the title should be “CONCLUSION”.

Line 603 – Again, explain why “may also provide superior benefit”. It is not clear.

Revision is necessary to improve the quality of English language. Examples: Line 35, Lines 38-39.

Author Response

ABSTRACT

  1. Lines 25-27 – This phrase is unnecessary and should be removed. The Abstract should begin at “This study…”

Ans: Thank you for reviewer’s suggestion. We have removed the sentence.

  1. Line 35 – Suggestion: Groups that received (instead of “Groups receiving”)

Ans: Thank you for reviewer’s great suggestion. We had change.

  1. Lines 38-39 – The phrase is unclear. Did the authors mean that all treatments suppressed the ACTL-induced effects (downregulation of C-II expression, and upregulation of MMP13, TNF-a and IL-1b expression)? Please, rewrite.

Ans: Thank you for reviewer carefully review. We have rewritten the sentence as the following” Furthermore, PSCC-II and unproteolyzed C-II suppressed ACLT-induced effects, such as the downregulation of C-II and upregulation of matrix metalloproteinase-13, tumor necrosis factor-α, and interleukin-1β.

  1. Line 40 – Why PSCC-II treatment “may provide superior benefits for OA management”? Either explain here or remove the phrase.

Ans: Thank you for reviewer’s great suggestion. We have rewritten the sentence as the following” These results indicate that PSCC-II has the protective effects of traditional undenatured C-II and provides superior benefits for OA management. These benefits encompass pain relief, anti-inflammatory effects, and the protection of cartilage and cancellous bone.” for explain superior benefits.

INTRODUCTION

  1. Lines 68 and 75 – “naïve” collagen – did the authors mean “native” collagen?

Ans: Thank you for reviewer’s suggestion. For more straightforward, we have changed “naïve” to “native”. The native collagen, meant without modification or removal of glycosylation on C-II, may lead to increased joint inflammation and T cell recruitment.

  1. Line 81 – What is “oral tolerance properties”?

Ans: Thank you for reviewer carefully review. Oral tolerance properties are an immune process the body uses to distinguish between innocuous compounds (e.g., dietary proteins, intestinal bacteria) and potentially harmful foreign invaders. For a clearer explanation, we have reedited the sentence: " Although undenatured C-II can be ingested orally without triggering an immune response. However, preserved posttranslational modifications or structures may trigger adverse immune reactions.”

RESULTS

  1. Lines 276-277 – Please, remove this phrase.

Ans: Thank you for reviewer carefully review. We have removed this phrase.

  1. Page 8, Figure 1 – “B” and “C” – it seems that the legend does not corresponds to the Figures. It seems that Figure 1-B is the mechanical allodynia, while Figure 1-C is swelling. Please, check.

Ans: Thank you for reviewer carefully review. We have checked and re-edited the Figure 1 Legend.

CONCLUSIONS

  1. Line 601 – There is only one Conclusion. So, the title should be “CONCLUSION”.

Ans: Thank you for reviewer carefully review. We have revised. (Line 620)

  1. Line 603 – Again, explain why “may also provide superior benefit”. It is not clear.

Ans: Thank you for reviewer carefully review. We have rewritten the sentence as the following” Our results conclude that PSCC-II has the protective effects of traditional undenatured C-II and provides superior benefits for OA management. These benefits encompass pain relief, anti-inflammatory effects, and the protection of cartilage and cancellous bone.”

Comments on the Quality of English Language

  1. Revision is necessary to improve the quality of English language. Examples: Line 35, Lines 38-39.

Ans: Thank you for reviewer carefully review. This manuscript had edited by Wallace Academic Editing (https://www.editing.tw/). The English editing certificate is the following:

Reviewer 2 Report

The paper titled “Oral Administration of Protease-Soluble Chicken Type II Collagen Ameliorates Anterior Cruciate Ligament Transection–Induced Osteoarthritis in Rats” (Manuscript ID nutrients-2554594) is well-written and very interesting. The Authors found that the oral administration of PSCC-II and undenatured C-II effectively reduced nociception sensitivity in rats with ACLT-induced OA.

1.     Is a similar type to oral protease-soluble chicken C-II (PSCC-II) found in other food sources of animal origin?

2.     What is the mechanism of decreased inflammations caused by this type of this collagen (e.g., reduced tumor necrosis factor-α or interleukin-1β)?

3.     May PSCC-II have a similar positive influence on human cartilage?

4.     Can PSCC-II be destroyed by metabolic animal or human processes, particularly during gastrointestinal passage (caused by enzyme destruction)?

Author Response

  1. Is a similar type to oral protease-soluble chicken C-II (PSCC-II) found in other food sources of animal origin?

Ans: We thank you for the reviewer great comments.

Currently, three configurations of C-II are used for nutraceutical applications in the prevention of OA: natural insoluble undenatured C-II, protease-soluble undenatured C-II, and hydrolyzed C-II. Natural insoluble undenatured C-II can be derived from chicken,[1] pig, [2] and salmon nasal cartilage, [3] and it is used as a nutraceutical ingredient under names such as UC-II (Lonza, Collavant of Bioiberica, S.A.U.) and SCP-II (Guzen Development). Protease (pepsin)-soluble undenatured C-II, such as EXT-II (Ryusendo), [3] is extracted from chicken sternal cartilage. C-II hydrolysate is primarily obtained from insoluble undenatured collagen fibrils from chicken sternal cartilage, such as in BioCell Collagen (Biocell Technology) [4] and KollaGen-II xs (Certified Nutraceuticals). [5] In PSCC-II, only pepsin-soluble undenatured C-II extracted from chicken sternal cartilage is available on the nutraceutical market. The use of marine pepsin-soluble undenatured C-II sourced from the skulls and cartilage of Nile tilapia and sturgeon is in the research stage. [5-7]

We also add the above sentences in the discussion of the revised manuscript.

Reference

  1. Bagi, C.M.; Berryman, E.R.; Teo, S.; Lane, N.E. Oral administration of undenatured native chicken type II collagen (UC-II) diminished deterioration of articular cartilage in a rat model of osteoarthritis (OA). Osteoarthritis Cartilage 2017, 25, 2080-2090, doi:10.1016/j.joca.2017.08.013.
  2. Di Cesare Mannelli, L.; Micheli, L.; Zanardelli, M.; Ghelardini, C. Low dose native type II collagen prevents pain in a rat osteoarthritis model. BMC Musculoskelet Disord 2013, 14, 228, doi:10.1186/1471-2474-14-228.
  3. Tomonaga, A.; Takahashi, T.; Tanaka, Y.T.; Tsuboi, M.; Ito, K.; Nagaoka, I. Evaluation of the effect of salmon nasal proteoglycan on biomarkers for cartilage metabolism in individuals with knee joint discomfort: A randomized double-blind placebo-controlled clinical study. Experimental and therapeutic medicine 2017, 14, 115-126, doi:10.3892/etm.2017.4454.
  4. Hector L Lopez, S.H., JE Sandrock, AW Kedia, TN Ziegenfuss. Effects of BioCell Collagen® on connective tissue protection and functional recovery from exercise in healthy adults: a pilot study. J Int Soc Sports Nutr 2014, P48.
  5. BS Bagatela, A.L., FL Affonso Fonseca, S Morton, J Gu, FF Perazzo1. Safety and Efficacy KollaGen II-xs: A 60-day Clinical Trial. West Indian Medical Journal 2016.
  6. Hou, C.; Li, N.; Liu, M.; Chen, J.; Elango, J.; Rahman, S.U.; Bao, B.; Wu, W. Therapeutic Effect of Nile Tilapia Type II Collagen on Rigidity in CD8(+) Cells by Alleviating Inflammation and Rheumatoid Arthritis in Rats by Oral Tolerance. Polymers 2022, 14, doi:10.3390/polym14071284.
  7. Luo, Q.B.; Chi, C.F.; Yang, F.; Zhao, Y.Q.; Wang, B. Physicochemical properties of acid- and pepsin-soluble collagens from the cartilage of Siberian sturgeon. Environmental science and pollution research international 2018, 25, 31427-31438, doi:10.1007/s11356-018-3147-z.

  1. What is the mechanism of decreased inflammations caused by this type of this collagen (e.g., reduced tumor necrosis factor-α or interleukin-1β)?

Ans: We thank you for reviewer great comments.

Studies have revealed that undenatured C-II offers joint protection through oral tolerance, wherein the immune system discerns harmless substances after oral ingestion.[1-3] Oral administration of undenatured chicken C-II can alleviate its nociception, reduce plasma levels of proinflammatory cytokines (TNF-α, IL-1b), and mitigate cartilage degradation, as evidenced by reduced plasma levels of C-telopeptide of C-II. [4] Extensive research has supported the protective effect of undenatured C-II against joint damage, which is primarily attributed to oral tolerance rather than its absorption or assimilation. Undenatured C-II helps counteract T cell–mediated inflammatory damage to C-II in joint cartilage. [5] The epitopes of native C-II cross the gut lumen through M cells and then transverse enterocytes. Payer’s patches (i.e., gut-associated lymphoid tissue, GALT) are gastrointestinal lymph nodes rich in T cells. Dendritic cells within GALT capture undenatured (i.e., native) C-II in its glycosylated form; repeated low-dose administration of undenatured C-II with its intact antigenic sites stimulates T regulatory cells (Tregs) and suppresses the activation of T helper (Th) cells. [6] Th cells develop “tolerance,” leading to a weakened immune response to cartilage collagen. This shift in immune balance leads to the dominance of Tregs and the suppression of anti-inflammatory cytokines (TNF-α, IL-1b), which are responsible for enhancing the activity of cartilage collagen–degrading MMP enzymes. [7] Therefore, endogenous collagen synthesis may be more effective in counteracting collagen destruction and inflammatory responses. Furthermore, amino acids and peptides derived from hydrolyzed collagens are absorbed into the systemic circulation and reach the articular cartilage. These compounds can stimulate the synthesis of extracellular matrix (ECM) macromolecules and encourage chondrogenic differentiation without inhibiting inflammation.[4] In the present study, protease M (Amano M)–soluble chicken C-II was classified as a form of PSCC-II with undenatured C-II. PSCC-II significantly inhibited the production of the proinflammatory cytokines IL-1b and TNF-α and ECM-degrading protein MMP13 in cartilage after ACLT (Figures 5P, 6H, and 6P). These proinflammatory cytokines inhibit the synthesis of major extracellular structural proteins, such as C-II and aggrecan. Concurrently, they induce the production of MMPs, disintegrin, and metalloproteinase with thrombospondin motifs by chondrocytes, contributing to the degradation of the ECM. We propose that protease M (Amano M)–processed PSCC-II may exhibit protective properties similar to those of native C-II in the context of OA. Moreover, the molecular weight of undenatured C-II was discovered to be approximately 300 kDa, whereas that of PSCC-II was 280–300 kDa. We also speculate that the telopeptides and glycosylated structures within the C-II fiber bundles of protease M (Amano M)–processed PSCC-II obtained through enzymatic proteolysis undergo alterations, potentially reducing autoimmunity. However, the precise protective mechanisms of PSSC-II in OA require additional investigation. Some of the aforementioned information has been incorporated into the Discussion section of the revised manuscript.

Reference

  1. Charriere, G.; Hartmann, D.J.; Vignon, E.; Ronziere, M.C.; Herbage, D.; Ville, G. Antibodies to types I, II, IX, and XI collagen in the serum of patients with rheumatic diseases. Arthritis Rheum 1988, 31, 325-332, doi:10.1002/art.1780310303.
  2. Bari, A.S.; Carter, S.D.; Bell, S.C.; Morgan, K.; Bennett, D. Anti-type II collagen antibody in naturally occurring canine joint diseases. Br J Rheumatol 1989, 28, 480-486, doi:10.1093/rheumatology/28.6.480.
  3. Cook, A.D.; Gray, R.; Ramshaw, J.; Mackay, I.R.; Rowley, M.J. Antibodies against the CB10 fragment of type II collagen in rheumatoid arthritis. Arthritis Res Ther 2004, 6, R477-483, doi:10.1186/ar1213.
  4. Martinez-Puig, D.; Costa-Larrion, E.; Rubio-Rodriguez, N.; Galvez-Martin, P. Collagen Supplementation for Joint Health: The Link between Composition and Scientific Knowledge. Nutrients 2023, 15, doi:10.3390/nu15061332.
  5. Bagchi, D.; Misner, B.; Bagchi, M.; Kothari, S.C.; Downs, B.W.; Fafard, R.D.; Preuss, H.G. Effects of orally administered undenatured type II collagen against arthritic inflammatory diseases: a mechanistic exploration. Int J Clin Pharmacol Res 2002, 22, 101-110.
  6. Park, K.S.; Park, M.J.; Cho, M.L.; Kwok, S.K.; Ju, J.H.; Ko, H.J.; Park, S.H.; Kim, H.Y. Type II collagen oral tolerance; mechanism and role in collagen-induced arthritis and rheumatoid arthritis. Mod Rheumatol 2009, 19, 581-589, doi:10.1007/s10165-009-0210-0.
  7. Lugo, J.P.; Saiyed, Z.M.; Lau, F.C.; Molina, J.P.; Pakdaman, M.N.; Shamie, A.N.; Udani, J.K. Undenatured type II collagen (UC-II(R)) for joint support: a randomized, double-blind, placebo-controlled study in healthy volunteers. J Int Soc Sports Nutr 2013, 10, 48, doi:10.1186/1550-2783-10-48.

  1. May PSCC-II have a similar positive influence on human cartilage?

Ans: We thank you for reviewer’s great comment.

NEXT-II, a form of pepsin-soluble chicken C-II, is a type of PSCC-II. In a 12-week open-label study involving healthy male and female participants, NEXT-II was administered in capsules at a daily dose of 40 mg. [1] That study demonstrated its efficacy in reducing joint pain and increasing knee flexibility and mobility. Furthermore, Western Ontario and McMaster Universities (WOMAC) osteoarthritis index visual analog scale scores indicated enhanced joint flexibility and mobility.[2] Studies with animal models have also evaluated the efficacy of NEXT-II for managing joint pain and inflammation. [3, 4] The safety of NEXT-II was demonstrated in a clinical study of healthy male and female participants in an open-label overdose trial. [5] Additionally, a randomized, double-blind, placebo-controlled, parallel-group study revealed that NEXT-II is efficacious in improving knee flexibility and mobility, reducing knee and low back pain, and enhancing motor function. [6] In the present study, pepsin- and protease M–processed PSCC-II reduced inflammation, alleviated nociceptive pain, and mitigated cartilage degradation in an ACLT-induced OA rat model. Thus, we suggest that protease M–processed the PSCC-II exerts a positive effect on human cartilage. This possibility merits future research.

Reference

  1. Yoshinari, O.; Marone, P.A.; Moriyama, H.; Bagchi, M.; Shiojima, Y. Safety and toxicological evaluation of a novel, water-soluble undenatured type II collagen. Toxicol Mech Methods 2013, 23, 491-499, doi:10.3109/15376516.2013.781255.
  2. Orie Yoshinari, H.M., Yoshiaki Shiojima, Hiromi Miyawaki. Evaluation of Efficacy and Safety of NEXT-II®, a Novel Water-Soluble, Undenatured Type II Collagen in Subjects with Potential Risks in the Knee Joint Health from Healthy Population. Functional Foods in Health and Disease 2015, 5, 14.
  3. Yoshinari, O.; Moriyama, H.; Shiojima, Y. An overview of a novel, water-soluble undenatured type II collagen (NEXT-II). J Am Coll Nutr 2015, 34, 255-262, doi:10.1080/07315724.2014.919541.
  4. Yoshinari, O.; Shiojima, Y.; Moriyama, H.; Shinozaki, J.; Nakane, T.; Masuda, K.; Bagchi, M. Water-soluble undenatured type II collagen ameliorates collagen-induced arthritis in mice. J Med Food 2013, 16, 1039-1045, doi:10.1089/jmf.2013.2911.
  5. Yoshiaki Shiojima, M.T., Ryohei Takahashi, Hiroyoshi Moriyama, Kazuo Maruyama, Debasis Bagchi, Manashi Bagchi. Safety of dietary undenatured type II collagen: a pilot open-label overdose clinical investigation. Functional Foods in Health and Disease 2022, 12, 13.
  6. Shiojima, Y.; Takahashi, M.; Takahashi, R.; Maruyama, K.; Moriyama, H.; Bagchi, D.; Bagchi, M.; Akanuma, M. Efficacy and Safety of Dietary Undenatured Type II Collagen on Joint and Motor Function in Healthy Volunteers: A Randomized, Double-Blind, Placebo-Controlled, Parallel-Group Study. J Am Nutr Assoc 2023, 42, 224-241, doi:10.1080/07315724.2021.2024466.

  1. Can PSCC-II be destroyed by metabolic animal or human processes, particularly during gastrointestinal passage (caused by enzyme destruction)?

Ans: We thank you for reviewer’s great comment.

Certain thick and short collagen fibers derived from insoluble undenatured C-II are undergo dissolution during gastric digestion, especially under low-pH conditions (pH 2.0). This process is accompanied by the release of soluble protein with a triple helix structure. The fibers of pepsin-processed undenatured C-II (one type of PSCC-II) exhibited mild thinning and curvature, and the triple helix structure remained nearly intact during passage through the gastrointestinal passage. That study asserted that the concentration of pepsin-processed undenatured C-II in digestive supernatants was 1.2 times (at pH = 2) to 12.4 times (at pH = 4) higher than that of native insoluble undenatured C-II, dependent on the pH.1 That finding suggests that pepsin-soluble undenatured C-II is likely to exert arthritis-relieving activity.[1] Thus, we propose that protease M (Amano M)–processed PSCC-II possesses properties similar to those of pepsin-processed PSCC-II during gastrointestinal transit. However, further investigations are required to confirm this.

Reference

  1. Xu, R.; Zheng, L.; Su, G.; Luo, D.; Lai, C.; Zhao, M. Protein solubility, secondary structure and microstructure changes in two types of undenatured type II collagen under different gastrointestinal digestion conditions. Food Chem 2021, 343, 128555, doi:10.1016/j.foodchem.2020.128555.
